# Effect of Lignin Plasticization on Physico-Mechanical Properties of Lignin/Poly(Lactic Acid) Composites

**DOI:** 10.3390/polym11122089

**Published:** 2019-12-13

**Authors:** Chan-Woo Park, Won-Jae Youe, Seok-Ju Kim, Song-Yi Han, Ji-Soo Park, Eun-Ah Lee, Gu-Joong Kwon, Yong-Sik Kim, Nam-Hun Kim, Seung-Hwan Lee

**Affiliations:** 1College of Forest & Environmental Science, Kangwon National University, Chuncheon 24341, Korea; chanwoo8973@kangwon.ac.kr (C.-W.P.); sngkgk@korea.kr (W.-J.Y.); songyi618@kangwon.ac.kr (S.-Y.H.); pojs04@kangwon.ac.kr (J.-S.P.); laa3158@kangwon.ac.kr (E.-A.L.); yongsikk@kangwon.ac.kr (Y.-S.K.);; 2Division of Wood Chemistry, National Institute of Forest Science, Seoul 02455, Korea; momoston@korea.kr; 3Kangwon Institute of Inclusive Technology, Kangwon National University, Chuncheon 24341, Korea; gjkwon@kangwon.ac.kr

**Keywords:** lignin composite, twin-screw extrusion, plasticization, coupling agent

## Abstract

Kraft lignin (KL) or plasticized KL (PKL)/poly(lactic acid) (PLA) composites, containing different lignin contents and with and without the coupling agent, were prepared in this study using twin-screw extrusion at 180 °C. Furthermore, ε-caprolactone and polymeric diphenylmethane diisocyanate (pMDI) were used as a plasticizer of KL and a coupling agent to improve interfacial adhesion, respectively. It was found that lignin plasticization improved lignin dispersibility in the PLA matrix and increased the melt flow index due to decrease in melt viscosity. The tensile strength of KL or PKL/PLA composites was found to decrease as the content of KL and PKL increased in the absence of pMDI, and increased due to pMDI addition. The existence of KL and PKL in the composites decreased the thermal degradation rate against the temperature and increased char residue. Furthermore, the diffusion coefficient of water in the composites was also found to decrease due to KL or PKL addition.

## 1. Introduction

Excessive use of petroleum-derived plastics has been known to cause serious environmental problems. Recently, issues regarding microplastic (consisting of plastic fragments of <5 mm) produced using non-biodegradable conventional plastics, such as polypropylene, polystyrene, and polyethylene, have been raised [1,2,3]. Microplastic is known to be an environmental contaminant causing soil and marine pollution which threatens agriculture and food production, and consequently human health [1,2,3]. As an alternative, many governments, global organizations, and multinational corporations have implemented policies to restrict the use of disposable plastic products. Additionally, development and utilization of bioplastics have lately been receiving attention [4,5,6]. Bioplastics can be classified into biomass-derived plastics and biodegradable plastics, which can include fossil-based plastics [7]. Poly(lactic acid) (PLA), polyhydroxyalkanoate (PHA), and thermoplastic starch (TPS) are representative biomass-derived bioplastics which possess biodegradability [6,7]. Poly(butylene succinate) (PBS) and polycaprolactone (PCL) are derived from petroleum resources but can be biodegradable [7,8,9]. 

PLA is considered one of the most promising biopolymers due to its excellent properties, which include biodegradability, biocompatibility, and renewability, as well as good mechanical properties [10,11,12]. PLA is derived from agricultural products (e.g., corn and potato) and is usually produced by ring-opening polymerization of lactide and condensation of lactic acid without polluting the environment during the production process [6,10,13]. However, despite its properties, the application of PLA has been limited because of its higher price and lower resistance to heat and water [11,13]. It has been proposed that compounding PLA with filler can improve its properties and remove some of its drawbacks [5,11,12,13,14,15]. Recently, some studies have introduced lignin to improve the performance of PLA and reduce the cost [10,13,14]. 

Lignin is one of the most abundant biopolymers, accounting for nearly 25% of lignocellulosic biomass [16,17,18,19]. It is usually easily obtained as a byproduct of the pulping industry and has some positive properties, including being biodegradable, non-toxic, and low-cost and having low density and excellent thermal and moisture resistance [20,21,22,23]. Due to these properties, research on lignin application for bioplastics has increased. Lignin has been known to have positive effects on composite properties. Some studies have observed that lignin addition enhances resistance for heat and moisture [14,15,24]. Furthermore, lignin has also been utilized as a stabilizer to prevent oxidation on plastic composites [25]. However, some studies have reported that the presence of lignin can deteriorate the mechanical properties of lignin-based composites. Lignin has been found to be incompatible with most aliphatic polyesters, including PLA, PBS, and PCL, thus deteriorating the mechanical properties of the composites [26,27,28]. However, it has been found that this strength deterioration, which is caused by lignin addition, can be overcome by adding coupling agents [26,29,30,31]. Isocyanate coupling agents, such as polymeric methylene diphenyl diisocyanate (pMDI), are known to form urethane linkages due to the reaction of the –NCO group in isocyanate coupling agents and the hydroxyl groups of lignin and aliphatic polyesters. This enhances the interfacial adhesion between lignin and polyesters [28,30,31,32]. In addition, dispersion of lignin in polyester matrices has been found to be significant in developing lignin-based high-performance polyester composites [26]. Plasticization is defined as the swelling effect in the matrix structure of lignin caused by plasticizers such as polyethylene glycol, triethylene glycol, lactic acid, and succinic acid [33,34]. Various plasticizers have been used to lower the glass transition temperature (*Tg*) of lignin to improve its thermal flowability [34]. Plasticizers increase the free volume by spacing polymer chains and the mobility of the chain segments, resulting in a decrease in the *Tg* and melt viscosity [35]. The lignin plasticization will also contribute to improving the dispersion of lignin in polymer matrices [26] and the processability and toughness of the thus-obtained composite. 

In this study, lignin/PLA composites were prepared using twin-screw extrusion. Furthermore, the effects of lignin plasticization and pMDI addition on tensile properties, melt flowability, and thermal and moisture stability of lignin/PLA composites were investigated.

## 2. Materials and Methods

### 2.1. Materials

For this study, PLA (Ingeo^TM^ Biopolymers 6400D) with a specific gravity of 1.24 and melt flow rate of 6.0 g/10 min at 210 °C and a load of 2.16 kg was obtained from Natureworks LLC (Minnetonka, MN, USA). Kraft lignin (KL) was isolated from the black liquor produced from hardwood chips by Moorim Pulp & Paper Co. (Ulsan, Korea) using the method described by Kim et al. [17]. The precipitated KL was recovered by filtration after adjusting the pH of the black liquor to below 2.0 using a concentrated HCl solution. This precipitate was repeatedly washed with fluent deionized water and later dried at 60 °C in an oven for 2 weeks. pMDI as a coupling agent with NCO content of 30.0–32.0% was obtained from Kumho Mitsui Chemicals Co. (Seoul, Korea).

### 2.2. Plasticization of KL

KL was mixed with ε-caprolactone (ε-CL) using a weight ratio of 80/20 (KL/ε-CL). The mixture was extruded at 150 °C with a screw rotation speed of 50 rpm using a twin-screw extruder (BA-11, Bautek Co., Pocheon, Korea) using an L/D ratio of 40/1 for the screw. The extrusion process was repeated thrice to prepare plasticized KL (PKL).

### 2.3. Preparation of KL or PKL/PLA Composites

KL and PKL were premixed with PLA pellets using weight ratios of 10/90, 20/80, and 30/70 (KL or PKL/PLA) and maintained in a vacuum dryer at 40 °C. To investigate the effect of a coupling agent, pMDIs of 1 and 3 phr, based on the total weight of KL or PKL/PLA mixtures, were added. Afterwards, KL or PKL/PLA composites were prepared using twin-screw extrusion at 180 °C, with a screw rotation speed of 50 rpm.

### 2.4. Morphology Observation

The fractured surface of the composites was coated with iridium using a high-vacuum sputter coater (EM ACE600, Leica Microsystems, Ltd., Wetzlar, Germany). The coating thickness was approximately 5 nm. The morphologies were observed using a scanning electron microscope (S-4800, Hitachi, Ltd., Tokyo, Japan) at the Central Laboratory of Kangwon National University.

### 2.5. Melt Flow Index (MFI) and Melt Viscosity Measurement

MFI and melt viscosity of the composites were measured with a melt flow indexer (MFI 4050, Rhopoint Instruments, Ltd., Hastings, UK). The samples were pre-heated in a vessel for 2 min at 180 °C, and MFI and melt viscosity were measured using a die of 2.09 mm diameter and a load of 2.16 kg. The measurement distance was set at 25.4 mm and MFI and melt viscosity were calculated using software for a melt flow indexer.

### 2.6. Tensile Test

First, the composites were hot-pressed at 180 °C for 1 min to form the sheets. Afterwards, the specimens were cut with a Type V sample cutter (KP-M2060, KIPAE E&T, Co., Ltd., Suwon, Korea) as described by the American Society for Testing and Materials D638 standard. Subsequently, they were maintained in a thermo-hygrostat at a relative humidity of 65% to remove the effects of relative humidity on the tensile properties. Tensile tests were conducted using a universal testing machine (H50K, Hounsfield Test Equipment, Redhill, UK) at a cross-head speed of 10 mm/min. A minimum of ten specimens of each sample were tested and the average values were reported.

### 2.7. FTIR Analysis

To investigate changes in the molecular structure of the coupling agent, FTIR analysis was conducted at the Central Laboratory of Kangwon National University using a Fourier transform infrared spectrophotometer (Frontier, PerkinElmer Inc., Daejeon, Korea) equipped with an attenuated total reflectance attachment. A total of 32 scans were run per sample in the range of 4000–500 cm^−1^.

### 2.8. Thermogravimetric Analysis (TGA)

TGA of the composites was performed using a thermogravimetric analyzer (Q2000, TA instruments Inc., New Castle, DE, USA) at the Central Laboratory of Kangwon National University. The samples (15–20 mg) were heated on a platinum pan under a nitrogen atmosphere. The range of scanning temperatures was 25–500 °C, with a heating rate of 10 °C/min. Derivative thermogravimetry (DTG) was performed by measuring mass loss with respect to temperature.

### 2.9. Water Absorption

The composites were hot-pressed at 180 °C for sheet formation and cut into rectangular shapes with dimensions of 30 mm × 0.5–0.6 mm × 30 mm (width × thickness × length). The specimens were dried at 60 °C in a vacuum dryer until the weight became constant and were then immersed in a water bath at 30 °C. The changes in weight were measured periodically using a balance, with a precision of 1 mg, until the moisture contents attained the equilibrium ($m_{\infty })$. The values of moisture content against the immersion time ($m_{t}$) were calculated using Equation (1), i.e.,
(1)$$m_{t}\left(\%\right)=\frac{W_{t}-W_{o}}{W_{o}}\times 100\left(\%\right)$$
where $W_{o}$ is the initial weight of the dried sample and $W_{t}$ is the sample weight against the immersion time in the water bath.

The diffusion coefficient (*D*) of the water into the composites was determined by Fick’s second law. For one-dimensional diffusions from the sheets, Equation (2) was used, i.e.,
(2)$$\frac{m_{t}}{m_{\infty }}=1-\frac{8}{\pi^{2}}\overset{\infty }{\underset{m=0}{\sum }}\frac{1}{{\left(2m+1\right)}^{2}}exp\left(-\frac{{\left(2m+1\right)}^{2}\pi^{2}Dt}{l^{2}}\right)$$
where *l* is the thickness of the composite sheet and *t* is the immersion time.

Equation (2) was applied to determine the diffusion coefficient for the long term (*D_2_*), where $m_{t}/m_{\infty }$ > 0.6. The diffusion coefficient for the short term (*D_1_*), where $m_{t}/m_{\infty }$ < 0.6, was determined using Equation (3), i.e.,
(3)$$\frac{m_{t}}{m_{\infty }}=4{\left(\frac{Dt}{\pi l^{2}}\right)}^{\frac{1}{2}}$$

## 3. Results and Discussion

Figure 1 shows the effects of lignin content and pMDI addition on morphological characteristics of the fractured surface of KL/PLA and PKL/PLA composites. Neat PLA shows smooth morphologies of the fractured surface. The separation between PLA and KL or PKL can be clearly observed in both KL/PLA and PKL/PLA samples without pMDI. It was observed that as the lignin content increased, the fractured surface became rougher. Furthermore, in the KL or PKL/PLA (30/70) sample without pMDI, lignin particles and many cavities caused by the removal of KL particles during fracturing were observed. Such morphologies might have resulted from a weak interfacial adhesion between the PLA matrix and KL or PKL due to their incompatibility. This weak interfacial adhesion can be overcome by adding pMDI to both the composites. The isocyanate group (–NCO) in pMDI can generate a linkage to hydroxyl groups in lignin and PLA during the extrusion process at 180 °C, resulting in the formation of the urethane linkage between lignin and PLA. Due to pMDI addition of 3 phr to the KL or PKL/PLA composites, the lignin particles and their pull-out cavities were rarely observed, showing a smooth fractured surface. The clear surface may indicate that urethane linkages between both the polymers, due to pMDI addition, can improve the intermolecular bonding. Sahoo et al. [28] prepared lignin/PBS (50/50) composites using a microextruder and investigated the effect of pMDI addition on the properties of PBS/lignin composites. It was found that 1% pMDI addition resulted in a urethane linkage between both the polymers due to a reaction between the –NCO group in pMDI and the –OH group of lignin and PBS. It was also reported that this linkage improved the intermolecular adhesion between lignin and PBS, thus improving the morphological characteristics of the composite.

Table 1 shows the effect of KL or PKL and pMDI addition on MFI and melt viscosity of KL/PLA and PKL/PLA composites. It was observed that increasing the KL and PKL contents increased and decreased the MFI and melt viscosity of both the composites, respectively. Kim et al. [13] have reported that 5% lignin addition to PLA decreases the shear and complex viscosities of the composite. PKL/PLA composites are known to have higher MFI and lower melt viscosity values than those of KL/PLA composites due to the good dispersibility of PKL in the PLA matrix. A previous study [26] has reported that plasticization of KL increases MFI in KL/PCL composites, indicating a decrease in melt viscosity. As pMDI was added to both the composites, MFI and melt viscosity decreased and increased, respectively. Park et al. [31] have studied the effect of pMDI addition on the properties of PBS/wood flour and PBS/wood flour/KL composites. It was noted by them that pMDI deteriorated the melt flowability of both the composites, indicating a decrease in MFI and increase in melt viscosity. The decrease in melt flowability due to pMDI addition might have been due to the coupling effect of urethane linkages between PLA and lignin. Yachon et al. [36] have investigated the effect of adding a coupling agent on the properties of PLA/poly(ether-b-amide) composites. It was also noted by these authors that a drastic decrease in melt flow rate occurred due to MDI addition. They reported that these results clearly indicated the occurrence of a coupling reaction.

The tensile strength and elastic modulus of neat PLA and KL or PKL/PLA composites are shown in Table 2. The tensile strength and elastic modulus of neat PLA were found to be 42 MPa and 2.1 GPa, respectively. The tensile strength of KL or PKL/PLA composites decreased with increasing content of KL and PKL. This might be due to weak interfacial adhesion between lignin and PLA in the composite, which has been reported in many studies [4,5,6,15,33]. PKL/PLA composites exhibited lower tensile strength than KL/PLA composites, while there was no significant difference in the elastic modulus. Su et al. [18] prepared alkali lignin/PLA films with glycerol and glutaraldehyde. They demonstrated that glycerol smoothened the surface and decreased the tensile strength. The plasticizer for lignin might have weakened the interaction of lignin molecules themselves, decreasing the reinforcement effect of lignin. This occurred despite the fact that the existence of the plasticizer improved lignin dispersibility in the polymer matrix. Due to pMDI addition in both the KL or PKL/PLA (30/70) composites, the tensile strength was increased, whereas the elastic modulus was maintained as a constant. During the twin-screw extrusion at 180 °C, urethane linkages were formed due to the reaction of the –NCO groups in pMDI and the hydroxyl groups in PLA and lignin, which improved the interfacial adhesion between PLA and lignin. In addition, urethane linkages have been shown to lead to hydrogen bonding between the N–H group of the urethane linkage and the carbonyl group in lignin and polyester [28]. Enhancement in the tensile strength was more noticeable in the PKL/PLA composites than that in the KL/PLA composites. The well-dispersed PKL in the PLA matrix might have formed a urethane linkage, resulting in better reinforcement of the PKL/PLA composites.

To confirm the effect of pMDI as a coupling agent in the composites, FTIR analysis was performed, and the obtained spectra are shown in Figure 2. In the spectrum of neat PLA, three bands at 1750, 1180, and 1080 cm^−1^, representing C=O, ester C–O, and carbonyl C–O stretches, respectively, are prominently visible. Peaks at 1620, 1520, and 1425 cm^−1^, which can be assigned to aromatic skeleton vibrations of phenyl propane, and at 1450 cm^−1^, corresponding to the C–H deformation combined with an aromatic ring vibration, were the main bands observed for a typical lignin structure. Furthermore, peaks at 1210, 1110, and 1030 cm^−1^ can be said to represent phenolic C–O stretching, aromatic C–H deformation in the plane, and C–O deformation in the methoxyl group, respectively, and can be seen to overlap with C–O stretch peaks in neat PLA. Additionally, pMDI showed a strong peak at 2250 cm^−1^ which corresponds to the isocyanate group (–N=C=O). However, this peak disappeared in the KL or PKL/PLA composites with pMDI, suggesting that the isocyanate group changed to a urethane group. The urethane group typically exhibited C–O, C–N, and C=O stretch vibrations at approximately 1200, 1600, and 1700 cm^−1^, respectively, which were found to overlap with the peaks of PLA and lignin.

Figure 3 shows the TGA and DTG curves for neat PLA and KL or PKL/PLA composites containing different lignin content, with and without pMDI. It was observed that thermal degradation of the neat PLA started at approximately 300 °C and was completely decomposed by 380 °C, with no residue remaining. KL or PKL/PLA composites indicated a similar tendency regarding thermal degradation. Thermal degradation of KL or PKL/PLA composites started at approximately 250 °C and later accelerated at 300 °C because of PLA decomposition. Due to 10% KL or PKL addition, the temperature at which thermal decomposition started was increased as a result of good thermal stability of lignin. However, it was observed that the weight of KL or PKL/PLA (30/70) composites started to slightly decrease at 250 °C. Although lignin has excellent thermal stability, the low molecular lignin was decomposed at 250 °C, which is lower than the initial temperature of thermal decomposition in neat PLA [16]. The char residue of the composites increased with increasing content of KL or PKL because lignin has aromatic structure frames that have the ability to form char [37]. The existence of char from lignin has been known to reduce the combustion heat and heat release rate of the composite, and, thus, some studies have suggested that lignin can be used as a flame-retardant additive in composites [12,15]. The PKL/PLA composites were found to have a lower amount of char residue than the KL/PLA composites because PKL contained 20 wt.% ε-CL, which could not generate char during thermal degradation. No significant effect was found on the thermal stability.

Figure 4 shows the water absorption behavior of neat PLA and KL or PKL/PLA composites containing different lignin content, with and without pMDI. The equilibrium moisture content at which no further water absorption occurred was determined as the maximum moisture absorption point. In all the samples, the water absorption amount increased rapidly for 2 h, followed by a very slow increase as the immersion time in water increased. The value of the maximum water absorption in neat PLA was found to be approximately 0.8%. The increase in KL and PKL contents in both the composites resulted in the increase of the value of maximum water absorption. Although lignin is an aromatic hydrophobic polymer consisting of a phenyl propane unit, a few hydroxyl groups of lignin might have caused the water absorption. The PKL/PLA composites were found to have a lower maximum water absorption than the KL/PLA composites. This could be because of possible weight loss due to ε-CL extraction by water from PKL during immersion, since ε-CL is a water-soluble monomer. Due to 3 phr pMDI addition, the water absorption against the immersion time and value of maximum water absorption in KL/PLA (30/70) were found to decrease, while no significant effect was observed on water absorption in PKL/PLA (30/70).

The diffusion coefficient of water was determined from the relationship between the water absorption amount and immersion time, as shown in Figure 4. Plots of $m_{t}$/$m_{\infty }$ against *t*^1/2^ and ln(1−$m_{t}$/$m_{\infty }$) against *t*, shown in Figure 5, were obtained using Equations (2) and (3), respectively. The diffusion coefficients for the short term (*D_s_*) and long term (*D_l_*) were determined from the gradient of linear regressions and the thus-obtained values are indicated in Table 3.

*D_s_* and *D_l_* of neat PLA were observed to be 9.2 × 10^12^/m^2^ s^−1^ and 5.0 × 10^12^/m^2^ s^−1^, respectively, indicating that the water absorption was quick in the initial stage but slowed down as the immersion time increased. Due to KL and PKL addition, both *D_s_* and *D_l_* in the composites decreased, while the maximum water absorption amount increased. The presence of lignin in the composites might have inhibited the rate of water absorption. Furthermore, as the KL content increased, the diffusion coefficients in both the short- and long-term stages decreased gradually. However, the PKL/PLA samples exhibited similar values of *D_s_* and *D_l_* despite the increase in PKL content, unlike *D_s_* and *D_l_* in the KL/PLA composites. pMDI addition caused an increase in the diffusion coefficient of the KL/PLA composites; however, in the PKL/PLA samples, no noticeable effect of pMDI addition was observed on the diffusion coefficients.

## 4. Conclusions

The effects of lignin plasticization and pMDI addition on the morphological characteristics, tensile properties, melt flowability, and thermal and water resistance properties of KL or PKL/PLA composites have been investigated in this study. KL or PKL/PLA composites were found to have rougher fractured surfaces with increasing KL and PKL contents; however, pMDI addition increased the compatibility between KL or PKL and PLA, resulting in a smoother fracture surface. An increase in compatibility between the polymers was also seen in the tensile properties of the composites. Deterioration in the tensile strength occurred in KL or PKL/PLA composites as the KL and PKL contents increased. However, the tensile strength was improved to some extent, since the urethane linkage formed from pMDI enhanced the interfacial adhesion between KL or PKL and PLA. Moreover, the influence of pMDI addition in improving the tensile strength in the PKL/PLA composites was found to be better than that for the KL/PLA composites. This was because the well-dispersed PKL in the PLA matrix might have increased the possibility of forming a urethane linkage. The existence of KL and PKL in the composites decreased the thermal degradation rate against the temperature and increased the char residues. As KL or PKL contents increased, the values of maximum water absorption in the composites were found to increase. KL or PKL/PLA samples were observed to have lower water diffusion coefficients for the short term and long term. These results will be useful as fundamental information when applying lignin to bioplastics.

## Figures and Tables

**Figure 1 polymers-11-02089-f001:**
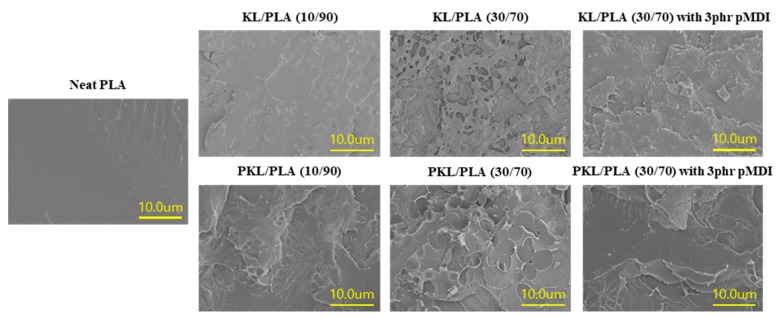
Scanning electron microscope micrograms of the fractured surface of neat poly(lactic acid) PLA, kraft lignin (KL)/PLA and plasticized KL (PKL)/PLA composites containing different content of lignin and polymeric diphenylmethane diisocyanate (pMDI).

**Figure 2 polymers-11-02089-f002:**
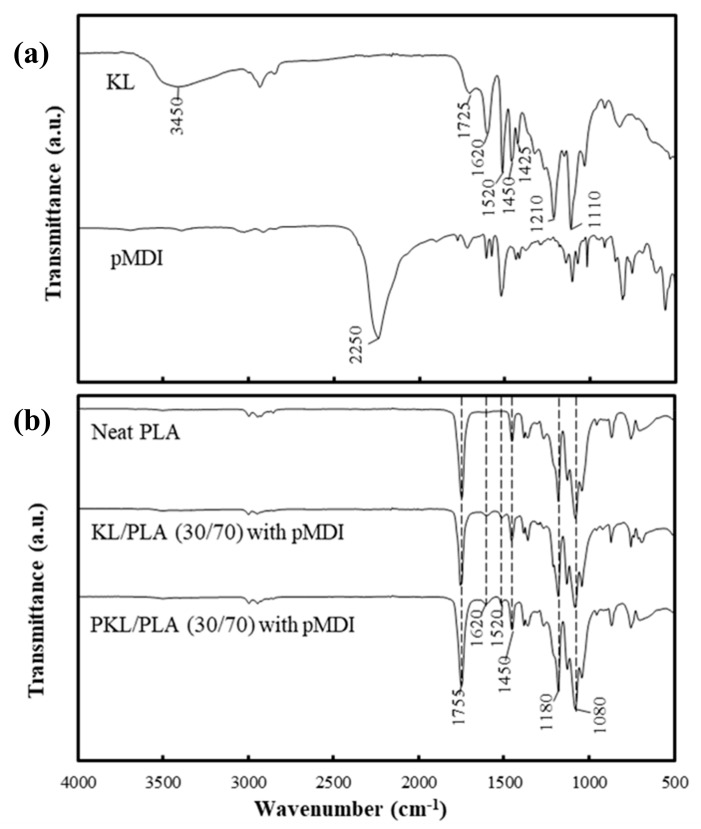
FTIR spectra of (**a**) neat PLA, pMDI, and (**b**) KL or PKL/PLA (30/70) composites, with and without pMDI.

**Figure 3 polymers-11-02089-f003:**
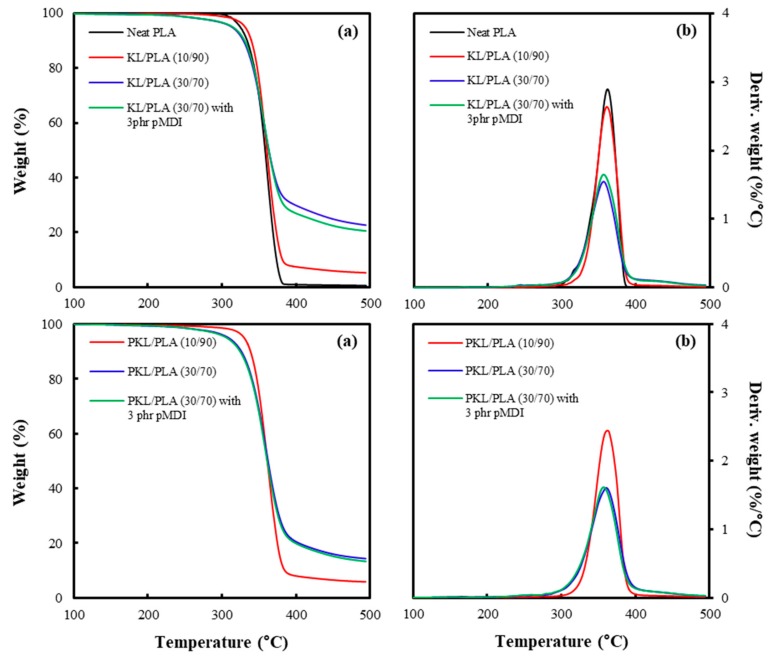
Thermogravimetric analysis (TGA) (**a**) and derivative thermogravimetry (DTG) (**b**) curves of neat PLA and KL or PKL/PLA composites containing different lignin content, with and without pMDI.

**Figure 4 polymers-11-02089-f004:**
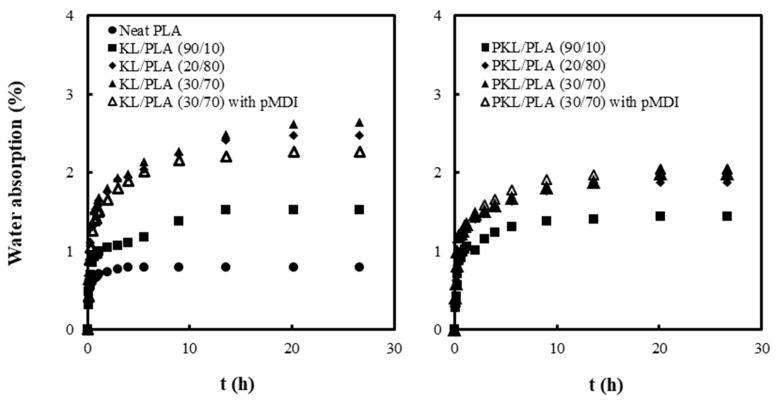
Water absorption curves of neat PLA and KL or PKL/PLA composites, with and without pMDI, with increasing immersion time in water.

**Figure 5 polymers-11-02089-f005:**
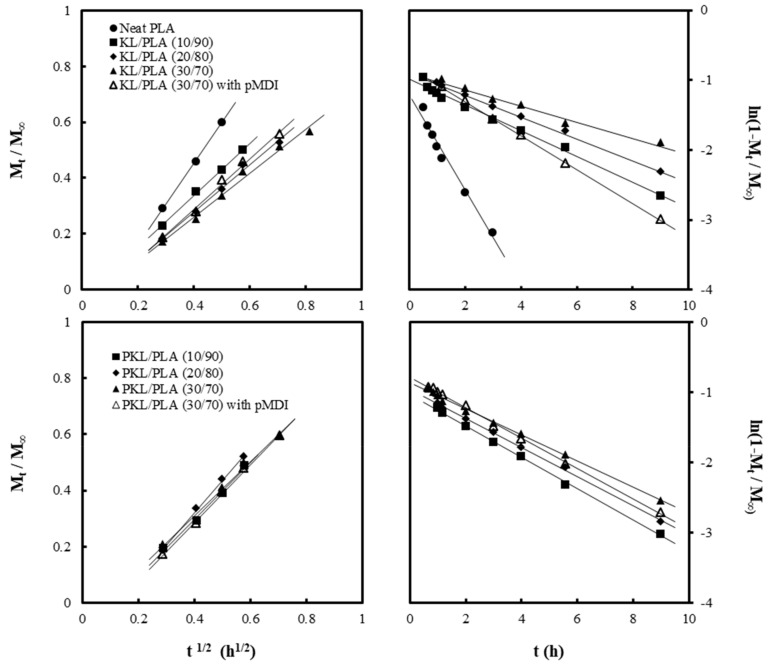
Plots of $m_{t}$/$m_{\infty }$ versus *t*^1/2^ and ln(1−$m_{t}$/$m_{\infty }$) versus *t* for neat PLA and KL or PKL/PLA composites containing different lignin content, with and without pMDI.

**Table 1 polymers-11-02089-t001:** Effect of KL or PKL and pMDI addition on melt flow index (MFI) and melt viscosity of KL/PLA and PKL/PLA composites.

KL or PKL Content (wt.%)		PLA Content (wt.%)	pMDI Content (phr)	Melt Flow Index (g/10 min)	Melt Viscosity (kPa·s)
KL	PKL				
-	-	100	-	2.2 ± 0.1	59.1 ± 3.7
10	-	90	-	6.0 ± 0.6	21.4 ± 2.0
20	-	80	-	20.9 ± 5.3	6.4 ± 1.8
30	-	70	-	26.4 ± 5.0	5.0 ± 1.0
30	-	70	1	19.7 ± 4.1	6.7 ± 1.6
30	-	70	3	10.6 ± 0.9	12.1 ± 1.0
-	10	90	-	66.0 ± 1.7	1.9 ± 0.2
-	20	80	-	159.7 ± 37.2	0.9 ± 0.4
-	30	70	-	280.3 ± 31.2	0.5 ± 0.2
-	30	70	1	127.2 ± 21.6	1.8 ± 1.2
-	30	70	3	91.9 ± 29.0	2.0 ± 0.9

**Table 2 polymers-11-02089-t002:** Effect of KL or PKL and pMDI content on tensile strength and elastic modulus of KL or PKL/PLA composites.

KL or PKL Content (wt.%)		PLA Content (wt.%)	pMDI Content (phr)	Tensile Strength (MPa)	Elastic Modulus (GPa)	Elongation at Break (%)
KL	PKL					
-	-	100	-	41.3 ± 3.1	2.1 ± 0.2	2.1 ± 0.1
10	-	90	-	27.0 ± 2.5	1.8 ± 0.1	1.6 ± 0.1
20	-	80	-	24.5 ± 1.4	1.7 ± 0.1	1.6 ± 0.1
30	-	70	-	21.8 ± 4.0	1.8 ± 0.1	1.6 ± 0.1
30	-	70	1	23.2 ± 2.5	1.7 ± 0.1	1.4 ± 0.2
30	-	70	3	25.3 ± 1.5	1.9 ± 0.1	1.4 ± 0.1
-	10	90	-	19.2 ± 3.8	1.8 ± 0.3	1.0 ± 0.2
-	20	80	-	18.6 ± 1.1	1.7 ± 0.1	0.8 ± 0.1
-	30	70	-	16.0 ± 2.5	1.7 ± 0.2	0.6 ± 0.1
-	30	70	1	19.6 ± 1.9	1.7 ± 0.1	0.6 ± 0.1
-	30	70	3	26.2 ± 3.0	1.7 ± 0.1	0.5 ± 0.1

**Table 3 polymers-11-02089-t003:** Maximum water absorption amount and diffusion coefficient of neat PLA and KL or PKL/PLA composites containing different lignin content, with and without pMDI.

Samples	Maximum Water Absorption Amount (%)	Diffusion Coefficient			
		Short-Term		Long-Term	
		D*_s_* × 10^12^/m^2^ s^−1^	R^2^	D*_l_* × 10^12^/m^2^ s^−1^	R^2^
**Neat PLA**	0.78	9.2	0.999	5.0	0.977
KL/PLA (10/90)	1.38	3.2	0.999	1.1	0.990
KL/PLA (20/80)	2.47	3.0	0.994	1.0	0.998
KL/PLA (30/70)	2.66	2.2	0.989	0.6	0.983
KL/PLA (30/70) with pMDI	2.27	3.7	0.992	1.5	0.999
PKL/PLA (10/90)	1.44	3.8	0.990	1.3	0.999
PKL/PLA (20/80)	1.87	4.0	0.892	1.1	0.999
PKL/PLA (30/70)	1.99	3.7	0.982	1.2	0.998
PKL/PLA (30/70) with pMDI	2.04	3.3	0.997	1.1	0.998

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
