# Peer review of "Effect of Lignin Plasticization on Physico-Mechanical Properties of Lignin/Poly(Lactic Acid) Composites"

_polymers, 2019, doi:10.3390/polym11122089_

Round 1

Reviewer 1 Report

The present paper  is certainly dealing with an open scientific and technical problems  and shos a strong point a good systematic experimentations , but suffer of two weak points which need to be improved  before the  possible publication. These are ;

A: after claiming the need of replace conventional efficient polymers with less efficient more biodegradable polymers , they use as coupling agent a very little environmentally accepted reagent ( diisocyanate).This last on the other side  produces probably  not reversible crosslinking ( see TGA) which  limits biodegradability.

b) The complex extruder feed is related simply to the material properties of the products without a molecular analysis of the possible  structural changes occurring during the extrusion as at that temperature several reactions could occur with consequences on the final biodegradability also not tested.

Some  detailed comments needing authors  consideration are also reported below:

Liness 38-39: poly(butylene succinate) (PBS), polycaprolactone (PCL), are not biopolymers but synthetic man made polymers which are biodegradable. Evan PLA is a man made polymer form a biomonomers. We shuld  be precise   particularly in the introduction directed even to non experts.

2.The experimental section carries the number 2 while subsection starts with 3. Please make correction

3.Information about properties of the starting materials in section 3.1 (should be 2.1) is very limited if any. Detailed data of commercial products should be reported.

4.When the effct of MDI as reactive compatibilized is reported at the starting of section 3 some insight about the  mechanism of interaction should be reported .

5-Figs 2 and 3 are  very difficult to read  and do not provide clear indications about the effects achieved.

6.Infrared spectra show that MDI has reacted but no evidence is reported  about the product formed .Estarction with some solvent should allow to separate different structure and demonstrate the real origin of properties variation

7,The TGA data are not clear and no explanation is given about the formation oa a residue in some cases.

8.NO data of elongation are reported to show the claimed plasticization effects. Also Tg data should help

9.A detailed analysis of the  possible reaction including text with simplified model system would help given the paper  better scientific value.

10.The English usage needs a substantiak improvement.

Author Response

Point 1: Liness 38-39: poly(butylene succinate) (PBS), polycaprolactone (PCL), are not biopolymers but synthetic man made polymers which are biodegradable. Evan PLA is a man made polymer form a biomonomers. We shuld  be precise  particularly in the introduction directed even to non experts.

Response 1: First of all, we thank for your critical advices. We agree with your comment that the PBS and PCL are biodegradable but synthetic polymers. For reader’s understanding we have been added to the Introduction as your advice in Line 39-43. Thank you.

Point 2: The experimental section carries the number 2 while subsection starts with 3. Please make correction

Response 2: Thank you. We’ve corrected the typos.

Point 3: Information about properties of the starting materials in section 3.1 (should be 2.1) is very limited if any. Detailed data of commercial products should be reported.

Response 3: Thank you for valuable comment. We’ve filled the detailed information about the commercial raw materials provided by the manufacture. Thank you.

Point 4: When the effect of MDI as reactive compatibilized is reported at the starting of section 3 some insight about the mechanism of interaction should be reported .

Response 4: Thank you for your review.  As your comment, we added the mechanism of the linkage between the polymers in line 159. Thank you.

Point 5: Figs 2 and 3 are very difficult to read  and do not provide clear indications about the effects achieved.

Response 5: Thank you for valuable review. We agree that the values in the graphs is less readable. We’ve tabulated them. Thank you.

Point 6: Infrared spectra show that MDI has reacted but no evidence is reported about the product formed .Estarction with some solvent should allow to separate different structure and demonstrate the real origin of properties variation

Response 6: Thank you for your critical comment. We’ve investigated the structural change of the pMDI to the urethane linkages by FT-IR. In the composites the isocyanate group (-N=C=O) at 2250 cm-1 was disappeared because it converts urethane groups. Actually, the urethane group should indicates the peaks on –NH (3300 cm-1) C=O(1700 cm-1), C-N(1600 cm-1), and C=C(1525 cm-1) but these peaks overlap the peaks in lignin. Therefore, we’ve confirmed the change in the structure of the pMDI by reducing the peak of isocyante group at 2250 cm-1. Thank you.

Point 7: The TGA data are not clear and no explanation is given about the formation oa a residue in some cases.

Response 7: As you know well, the lignin has aromatic structure with high carbon content, thus, can convert char over temperature of 400℃.  As the lignin content increased, the char residues were also increased. As your comment, we’ve described the reason for the variation in the char residue in line 245. Thank you.

Point 8: NO data of elongation are reported to show the claimed plasticization effects. Also Tg data should help

Response 8: Thank you for valuable review. We’ve added the elongation data of the composites to the table summarizing the tensile strength and elastic modulus. We agree on the need of for TG data. Actually, we’re preparing a new paper on the thermal properties of these composite used in this paper. We’ve obtained good results by DSC analysis, for example, variation of Tg, isothermal crystallinity, and non-isothermal crystallinity in the composites. Therefore, Tg data will be presented with further consideration in another paper. Thank you.

Point 9: A detailed analysis of the possible reaction including text with simplified model system would help given the paper  better scientific value.

Response 9: Thank you for your valuable advice. The urethane linkages as a result of the reaction isocyanate groups in pMDI with the hydroxyl groups in PLA and lignin have been drawn in the scheme. Then, we’ve added the scheme as graphical abstract. Thank you.

Point 10: The English usage needs a substantiak improvement.

Response 10: This manuscript was already inspected by native English speakers before the submission. Another native speaker has examined and revised this manuscript again. Thank you.

Reviewer 2 Report

This paper is about the effect of lignin plasticization on lignin/poly composites. Based on my view, so far this paper doesn't meet the journal requirement. I have the following specific comments:

1.The novelty of the work should be strengthened in introduction.

2.What is the basis of the preparation parameter of lignin/poly (lactic acid) system? How to determine the optimum performance of this system.

3. Does the source of KL, PKL and PLA influence the test results? aka. production method, particle size, chemical components. The authors may need to include related discussion in the paper.

4. Please improve the quality of Figure 1, as the included text is hardly readible. Please also do this on other figures.

5. The paper could be enhanced by discussing how the findings could be implemented.

6. The objective of this research is not very clear. In addition, what are specific conclusions and recommendations? Do the authors recommend this composite? What are the applicable occasions? Please make those more clear.

7. There are a few typos here and there through the paper (not that many though). Standard English technical writing is mandated.

8. The conclusion part should be modified. It should be more concise and highly concluded. Don't just summarize the test reseults.

Author Response

Point 1: The novelty of the work should be strengthened in introduction.

Response 1: First of all, we thank for your critical advices. In this manuscript, we’ve proposed the plasticization of lignin in order to improve the physical and mechanical properties of lignin based composite. We’ve supplemented the effect of lignin plasticization on properties of lignin based composites as well as properties of plasticized lignin in the Introduction in line 71. Thank you.

Point 2: What is the basis of the preparation parameter of lignin/poly (lactic acid) system? How to determine the optimum performance of this system.

Response 2: Thank you for your comment. In this study, we’ve prepared lignin-PLA composite via twin-screw extrusion. Actually the temperature in the vessel, rotational speed of the screws can affect the properties of the obtained composite. We already established the best condition for preparation of lignin composite via twin-screw extrusion in previous study. Thus, we fixed the temperature and rotational speed of screws at best conditions, and varied the content of lignin and plasticized lignin. Thank you.

Point 3: Does the source of KL, PKL and PLA influence the test results? aka. production method, particle size, chemical components. The authors may need to include related discussion in the paper.

Response 3: As u know well, the lignin has different properties, dependent on isolation method for lignin, and is classified into kraft lignin, alkali lignin, and so on. Plasticized lignin also differs in properties such as morphology, TGA and Tg, dependent on the type of plasticizer. However, in this study, we’ve used only kraft lignin because it is the most commercial lignin obtained pulping process. Moreover, we’ve ever prepared plasticized lignins using various polyols such PEG. If we used various plasticized lignins for this research, we could have shown lots of results on effect of plasticizers but we selected just one of the best plasticized lignin because the plasticized lignin with ε-caprolatone was better than others plasticized lignins. And, we’ve focused on the effect of lignin plasticization and pMDI addition on the composites, not the effect of the type of lignin and plasticizers. Thank you so much.

Point 4: Please improve the quality of Figure 1, as the included text is hardly readible. Please also do this on other figures.

Response 4: We’ve modified the Figure 1 to make it easier to read. Thank you.

Point 5: The paper could be enhanced by discussing how the findings could be implemented.

Response 5: Thank you for your valuable comment. In this study, we’ve found that the lignin plasticization and coupling agent can improve the strength and melt flowability of lignin composites. Thus, we have plans to use the obtained composite as 3D printing materials. Thank you.

Point 6: The objective of this research is not very clear. In addition, what are specific conclusions and recommendations? Do the authors recommend this composite? What are the applicable occasions? Please make those more clear.

Response 6: In order to make our research objectives clearer, we’ve created a graphical abstract and added in the manuscript. We’ve expected that the scheme on the pMDI reactions with PLA and lignin will help readers better understand the goals of our research. Thank you.

Point 7: There are a few typos here and there through the paper (not that many though). Standard English technical writing is mandated.

Response 7: We’ve reviewed and checked this manuscript with the help of native English speakers. Thank you very much.

Point 8\:. The conclusion part should be modified. It should be more concise and highly concluded. Don't just summarize the test reseults.

Response 8: Thank you for your critical comment. We’ve investigated the basic properties of the lignin-based composite prepared from plasticization lignin, and effect of pMDI addition on the properties of the composites. We’ve expected that these fundamental results can contribute to apply lignin to bioplastic industry. Thus, the we’ve added the importance of basic research on lignin-based plastic in Conclusion. Thank you.

Reviewer 3 Report

In my opinion it is an interesting article about the mechanical chracteristics of PLA composites reinforced with pure and plasticised Kraft Lignin. It can be printed without correctoins.
Only one typing error should be corrected:

- page 4 / line 153: 'week' --> 'weak'

Author Response

Point 1: page 4 / line 153: 'week' --> 'weak'

Response 1: We thank for you review. We’ve modified the error. Thank you for helping to improve the quality of the paper.

Round 2

Reviewer 1 Report

I have appreciated relevant impprovements reported  and  convincing reaction to my comments in the revised ms. It can be accepetd on my side.

Reviewer 2 Report

The quality of this paper has been obviously improved. I think my previous comments have been well addressed. Therefore, this paper is acceptable now based on my view.